# Real-World Experience among Elderly Metastatic Breast Cancer Patients Treated with CDK4/6 Inhibitor-Based Therapy

**DOI:** 10.3390/cancers16091749

**Published:** 2024-04-30

**Authors:** Thomas N. O’Connor, Emily Schultz, Jianxin Wang, Tracey O’Connor, Ellis Levine, Erik S. Knudsen, Agnieszka K. Witkiewicz

**Affiliations:** 1Department of Molecular and Cellular Biology, Roswell Park Comprehensive Cancer Center, Buffalo, NY 14263, USAemily.schiller@roswellpark.org (E.S.); jianxin.wang@roswellpark.org (J.W.); 2Department of Medicine, Roswell Park Comprehensive Cancer Center, Buffalo, NY 14263, USA; tracey.oconnor@moffitt.org (T.O.); ellis.levine@roswellpark.org (E.L.); 3Department of Pathology, Roswell Park Comprehensive Cancer Center, Buffalo, NY 14263, USA

**Keywords:** breast cancer, elderly, CDK4/6 inhibitor, metastatic

## Abstract

**Simple Summary:**

Although women 70 years of age and older represent a large percentage of newly diagnosed breast cancer cases, older individuals are drastically underrepresented in clinical trials. Most women diagnosed with breast cancer at 70 years of age and older present with hormone receptor-positive (HR+) breast cancer. Cyclin-dependent kinase (CDK) 4/6 inhibitor-based therapy has become the standard of care for metastatic HR+ breast cancer. This study reports that metastatic breast cancer patients ≥70 years of age receiving CDK4/6 inhibitor-based treatment, particularly when combined with an aromatase inhibitor, experience prolonged progression-free survival but display no such prolongation in overall survival and report more frequent adverse effects to treatment. The reported findings can aid in the development of optimized treatments based on age and help older patients weigh treatment options.

**Abstract:**

The largest portion of breast cancer patients diagnosed after 70 years of age present with hormone receptor-positive (HR+) breast cancer subtypes. Cyclin-dependent kinase (CDK) 4/6 inhibitor treatment, in conjunction with endocrine therapy, has become standard-of-care for metastatic HR+ breast cancer. In total, 320 patients with metastatic breast cancer receiving CDK4/6 inhibitor combined with fulvestrant or an aromatase inhibitor were enrolled in an ongoing observational study or were included in an IRB-approved retrospective study. All patients receiving CDK4/6 inhibitor-based therapy that were ≥70 years of age (n = 111) displayed prolonged progression-free survival (27.6 months) as compared to patients <70 years of age (n = 209, 21.1 months, HR = 1.38, *p* < 0.05). Specifically, patients receiving a CDK4/6 inhibitor with an aromatase inhibitor who were ≥70 years of age (n = 79) displayed exceptionally prolonged progression-free survival (46.0 months) as compared to patients receiving the same treatment who were <70 years of age (n = 161, 21.8 months, HR = 1.71, *p* < 0.01). However, patients ≥70 years of age also experienced more frequent adverse responses to CDK4/6 inhibitor-based treatment leading to dose reduction, hold, or discontinuation than the younger cohort (69% and 53%, respectively). Treatment strategies that may decrease toxicity without affecting efficacy (such as dose titration) are worth further exploration.

## 1. Introduction

Biomarkers are universally used to inform treatment for various cancer types. Hormone receptor (HR) status and human epidermal growth factor receptor 2 (HER2) status are useful in determining the initial treatment strategy for women with metastatic breast cancer. In HR+, HER2− clinical presentations, endocrine therapy combined with cyclin-dependent kinase 4/6 (CDK4/6) inhibitor therapy has become the standard of care [1,2,3,4,5,6]. This treatment strategy has also been implemented in other breast cancer subtypes with varying success [7]. The dependence of breast cancer on CDK4/6 and its binding partner cyclin D1 for progression through the G1/S phase of the cell cycle has made breast cancer a prime target for the implementation of CDK4/6 inhibitor-based therapy [8]. However, RB1 loss, amplification of CDK4/6/cyclin D, and amplification of CDK2/cyclin E, among many other mechanisms of resistance, have resulted in CDK4/6 inhibitor-based therapies being effective but not curative for the treatment of metastatic breast cancer [9,10,11,12,13,14,15].

One limitation of breast cancer clinical trials is the underrepresentation of elderly populations in such studies [16,17,18,19]. This is despite the fact that elderly populations represent a large percentage of the newly diagnosed cases, with the majority of such cases being HR+ and HER2− presentations [20,21]. Recently, there has been an increased focus on specifically addressing the needs and outcomes of older breast cancer populations [22,23,24,25,26,27,28], with the European Society of Breast Cancer Specialists and the International Society of Geriatric Oncology recommending 70 years of age as an appropriate stratification point [29].

Palbociclib, ribociclib, and abemaciclib are the currently FDA-approved CDK4/6 inhibitors indicated for use in combination with endocrine therapy in the form of an aromatase inhibitor (AI) such as anastrozole, letrozole, and exemestane, or the selective estrogen receptor degrader fulvestrant (FUL) for the treatment of metastatic breast cancer. In the corresponding clinical trials of palbociclib (PALOMA-2 and 3) [30,31], ribociclib (MONALEESA-2, 3, and 7) [32,33,34,35], and abemaciclib (MONARCH-2 and 3) [36,37], all three drugs in combination with endocrine therapy were found to be safe and effective in extending the duration of progression-free survival (PFS) when compared to endocrine therapy alone across young and old patient populations. 

While most studies have found that older populations tend to experience more adverse effects and consequently more dose reductions, holds, or discontinuations from CDK4/6 inhibitor-based therapy, there has been significant variability in the extent to which older populations benefit from CDK4/6 inhibitor-based therapy as compared specifically to younger patients. Thus, despite the relative safety and efficacy of CDK4/6 inhibitor-based therapy as compared to endocrine therapy alone, continued stratification based on age to assess the response of older individuals to CDK4/6 inhibitor-based therapy is not only warranted but required to effectively counsel patients and guide treatment decisions. Based on several previous studies that have shown significant differences in drug metabolism, efficacy, and adverse responses based on age and endocrine therapy [38,39], this study assessed the real-world response to therapy at a single cancer center (Roswell Park Comprehensive Cancer Center, RPCCC) as stratified by age (70 years) and the type of endocrine therapy used (AI or FUL). 

## 2. Materials and Methods

### 2.1. Patient Selection

A chart review was performed to determine study eligibility for over 4000 patients being seen in the Breast Cancer Clinic of RPCCC. Patients ≥ 18 years of age with stage IV metastatic HR+/HER2− breast cancer were considered eligible if treated with a CDK4/6 inhibitor (either palbociclib, abemaciclib, or ribociclib) and endocrine therapy after being diagnosed with metastatic disease between July 2020 and April 2024. A total of 320 eligible patients taking an aromatase inhibitor (AI, either letrozole, exemestane, or anastrozole) or fulvestrant (FUL) were included in this study from two separate protocols: 69 patients from a retrospective chart review protocol and 251 newly enrolled patients consented to a combined retrospective and prospective chart review protocol (NCT04526587) [40]. Both protocols were approved by the RPCCC Institutional Review Board (IRB) and were conducted in accordance with the Declaration of Helsinki.

### 2.2. Tissue Selection and Subtyping

Surgical pathology slides from the RPCCC Department of Pathology were reviewed for all study patients to ensure tumor adequacy for any procedure occurring before progressing to CDK-based therapy. Chosen cases were sectioned and a breast pathologist (AKW) outlined areas of the highest tumor cellularity. Slides were sent for RNA sequencing using HTG EdgeSeq technology’s Oncology Biomarker Panel at HTG Molecular Diagnostics, Inc. The AIMS [41] R package was then used on the normalized sequencing data to predict intrinsic cancer subtypes for each sample.

### 2.3. Statistical Analysis

The endpoint of interest, progression-free survival (PFS), was calculated as the time from initiation of CDK4/6 inhibitor to scan or biopsy-proven progression, as determined by RPCCC clinicians. Overall survival (OS) was also reported and was calculated as the time from treatment initiation to death, in months. The duration of median PFS and OS was determined using Kaplan–Meier curves and patient groups were compared using the log-rank test. In this study, patients were placed into subsets based on age at CDK initiation (stratified at 70 based on updated recommendations from the International Society of Geriatric Oncology for conducting research on older breast cancer patients [29]) as well as whether they were taking AI or FUL in combination with CDK4/6 inhibitor therapy. Adverse effects of treatment were self-reported and dose reductions, holds, and treatment discontinuations were conducted as per standard protocols. Various clinical and pathological features were tested in a univariate analysis using the Cox proportional hazards model in each group of patients to determine hazard ratios and 95% confidence intervals. For METABRIC analyses, the METABRIC dataset was downloaded from cbioportal (https://www.cbioportal.org/study/summary?id=brca_metabric), accessed on 29 February 2024. This dataset contained expression data of 20637 genes for 1980 samples. The HR+/HER2− subset of data was extracted by setting ER_STATUS equals “positive” and HER2_STATUS equals “negative” in a custom R code. This resulted in 1398 patient samples in the selected HR+/HER2− dataset. To compare the survival difference between the age groups in this dataset, patients were stratified into younger than 70 years of age (AGE_AT_DIAGNOSIS < 70) and older than or equal to 70 years of age. All analyses and figures were run using R software v4.3.2 with the survival, survminer, and ggplot2 packages.

## 3. Results

### 3.1. Study Design

A total of 320 patients receiving CDK4/6 inhibitor-based treatment for metastatic breast cancer were included in this observational/retrospective study (Figure 1). Due to the known differences in duration of PFS in individuals treated with either AI or FUL [38,42], the cohort was subdivided as such, with 240 patients receiving CDK4/6 inhibitor plus AI and 80 patients receiving CDK4/6 inhibitor plus FUL. The cohorts were further stratified by age with a cutoff of 70 years of age. This age was selected in part due to the significant reduction in drug metabolism after 70 years of age [39]. In the CDK4/6 inhibitor plus AI cohort, there were 79 individuals ≥70 years of age and 161 individuals <70 years of age. In the CDK4/6 inhibitor plus FUL cohort, there were 32 individuals ≥70 years of age and 48 individuals <70 years of age. 

HTG EdgeSeq analysis on patient biopsies collected before initiation of treatment was used to report the prevalence of each breast cancer subtype in the four cohorts assessed (Table 1). While most patients receiving CDK4/6 inhibitor-based therapy presented with HR+ subtypes (luminal A, luminal B, and normal), the prevalence of HR− subtypes ranged from 5.7% to 25.0% across the four cohorts. Interestingly, in the AI-treated group, the <70 cohort displayed a similar percentage of patients presenting with the Luminal A and luminal B subtypes (45.7% and 48.6%, respectively) whereas the ≥70 cohort displayed a higher percentage of patients presenting with the luminal A subtype (64.4%) as compared to luminal B (28.9%) (*p*-value = 0.06).

### 3.2. Clinicopathological Variates

Scarff–Bloom–Richardson (SBR) breast cancer scoring is a widely accepted prognostic tool for grading disease on a scale of 1–3, with higher values being indicative of more aggressive tumors [43]. The SBR grading system considers gland/tubule formation, nuclear pleomorphism, and mitotic count. Of note, the <70 cohort displayed a higher prevalence of SBR3 presentations (28.2%) as compared to the ≥70 cohort (16.2%), although the only significantly different contributor to the duration of PFS in the different aged cohorts observed in our dataset was SBR2 in the AI-treated group (Table 2). The ≥70 cohort displayed a higher prevalence of patients presenting with visceral (55.9%), distant (99.1%), and recurrent metastases (73.9%), as compared to the <70 cohort (43.1%, 93.3%, and 64.6%, respectively) and all three variates displayed a significantly different contribution to PFS between the two age groups within the AI-treated group (Table 2). Additionally, HER2 1+, progesterone receptor-positive (PR+), and no prior endocrine therapy (ET) also displayed significantly different contributions to PFS between the two age groups within the AI-treated group (Table 2).

### 3.3. Survival Based on Endocrine Therapy

As expected, the AI and FUL-treated groups displayed different durations of PFS. In the combined cohort irrespective of age, the AI-treated group displayed a median PFS of 26.7 months, whereas the FUL-treated group displayed a median PFS of 16.6 months (hazard ratio, HR = 1.99, *p* = 1 × 10^−5^) (Figure 2a). In the <70 cohort, the difference in duration of median PFS was smaller at 21.83 months for the AI-treated group and 17.16 months (HR = 1.59, *p* = 0.01) for the FUL-treated group (Figure 2b). However, in the ≥70 cohort, the difference in median PFS was far larger with a median PFS of 45.96 months for the AI-treated group and 15.02 months for the FUL-treated group (HR = 3.31, *p* = 1 × 10^−5^) (Figure 2c). A similar trend was observed in the ≥70 cohort for overall survival (OS) as well, as the AI-treated group displayed an OS of 67.20 months as compared to 31.43 months for the FUL-treated group (HR = 2.57, *p* = 0.003) (Appendix A).

### 3.4. Survival Based on Age

The data were further analyzed to directly assess the impact of age on PFS. In the combined AI and FUL-treated group, the ≥70 cohort displayed a median PFS of 27.58 months, whereas the <70 cohort displayed a median PFS of 21.11 months (HR = 1.38, *p* = 0.049) (Figure 3a). Interestingly, no difference in PFS within the same age stratification at 70 was observed in the METABRIC database (Appendix A). This difference in median PFS by age was even greater in the AI-treated group, as the ≥70 cohort displayed a median PFS of 45.96 months and the <70 cohort displayed a median PFS of 21.83 months (HR = 1.71, *p* = 0.01) (Figure 3b). However, age had no impact on OS in the AI-treated group (Appendix A). In the FUL-treated group, there was no significant difference observed in the median duration of PFS between the <70 (17.16 months) and ≥70 (15.02 months) cohorts (Figure 3c) but there was a significant difference in OS between the <70 (66.38 months) and ≥70 (31.43 months) (HR = 0.49, *p* = 0.03) cohorts (Appendix A). Similarly, OS was observed to be significantly shorter in patients ≥70 as compared to patients <70 from the METABRIC database [44] (Appendix A).

### 3.5. Adverse Effects

Although the ≥70 cohort on average, and particularly in the AI-treated group, displayed prolonged median duration of PFS, the <70 cohort on average had fewer reported adverse effects to treatment, consistent with previous studies [20,23,45]. Regardless of AI or FUL treatment, the ≥70 cohort reported a higher percentage of patients that experienced fatigue, diarrhea, thrombocytopenia, anemia, and/or an adverse effect that led to a dose reduction, hold of treatment, or discontinuation (Table 3).

## 4. Discussion

Here, we report real-world findings from a single cancer center regarding the treatment response of patients presenting with metastatic breast cancer to CDK4/6 inhibitor-based therapy, assessing both treatment benefits, as measured by PFS and OS, and adverse effects, with a focus on disparities between individuals less than or at least 70 years of age. Both the desired and adverse responses to treatment in the ≥70 cohort may be due in part to the discrepancy of drug metabolism between older and younger patients, with the age 70 cutoff emerging as an effective stratification point [29,39]. With reduced cytochrome P450 activity and slowed metabolic degradation, it is feasible to assume a prolonged duration of action, both on and off-target, in the older cohort. Thus, treatment decisions for older patients should be made on a case-by-case basis, carefully weighing both the benefits and adverse effects.

One limitation of the current study was the lack of a control cohort receiving endocrine therapy alone to enable a comparison of the magnitude of the benefit of CDK4/6 inhibitor inclusion in the treatment regimen within the different age cohorts. However, extensive prior research has been conducted validating the efficacy of the CDK4/6 inhibitors palbociclib, ribociclib, and abemacicilib to extend PFS duration (and abemaciclib and ribociclib to extend OS duration) when combined with endocrine therapy, as compared to endocrine therapy alone, for the treatment of HR+ and HER2− metastatic breast cancer in pre-, peri-, and postmenopausal women [4,46,47]. Similar to previous studies [23,45], it was concluded in the current study that patients ≥70 experience prolonged PFS in response to CDK4/6 inhibitor-based therapy, particularly when combined with AI. Although, notably, there have been studies that contradict this finding, noting either no difference in PFS between patients <70 and patients ≥70 receiving CDK4/6 inhibitor-based therapy [25] and even studies reporting the opposite effect of reduced PFS in the aged cohort [20].

While our analyses report a similar efficacy across age groups in terms of PFS in the FUL-treated cohort, there is a profound difference in the benefit of response to treatment with CDK4/6 inhibitor plus AI between the two age groups assessed at 45.96 vs. 21.83 months median PFS for the ≥70 and <70 cohorts, respectively (Figure 3). In this study, we do observe that the older cohort displays a reduced prevalence of clinical presentations characterized as SBR3 (16.2%), as compared to the younger cohort (28.2%). Similar to the observed differences in median PFS in the AI and FUL-treated groups, the SBR3 prevalence per age group follows the same trend, with a larger gap in the AI group at 16.5% and 29.8% in the ≥70 and <70 cohorts, respectively, and a smaller gap in the FUL group at 15.6% and 22.9% in the ≥70 and <70 cohorts, respectively (Table 2). However, several factors associated with worse disease progression outcomes contradict the observed difference in PFS across the age groups, as the ≥70 cohort displayed a higher percentage of individuals presenting with visceral tumors (55.9%), distant metastases (99.1%), and having received prior endocrine therapy (68.5%). This is in comparison to the <70 cohort that displayed 43.1% of individuals presenting with visceral tumors, 93.3% of individuals presenting with distant metastases, and 63.6% of individuals receiving prior endocrine therapy (Table 2). Future studies are necessary to fully extrapolate the baseline characteristics that contribute to this age discrepancy in the overall benefit of CDK4/6 inhibitor-based therapy.

In terms of OS duration and adverse effects of CDK4/6 inhibitor-based therapy, there is far more agreement across several prior studies and the current study when comparing older and younger cohorts. In agreement with previous findings [28,45], the current study reports that there is no significant difference in OS between patients <70 and patients ≥70 receiving CDK4/6 inhibitor-based therapy either in the combined cohort or in the AI-treated cohort and that the older population experiences reduced OS in the FUL-treated cohort (Appendix A). 

Similarly, consistent with most prior studies [20,23,45], the current study reports more prevalent adverse effects in response to CDK4/6 inhibitor-based treatment in patients ≥70 compared to patients <70, leading to more frequent dose reductions, holds, and discontinuations in the older cohort with the most prevalent adverse effects experienced including neutropenia, fatigue, and diarrhea. Of note, more than half of the ≥70 cohort experienced fatigue and over a quarter of the ≥70 cohort experienced diarrhea in response to CDK4/6-based treatment. Additionally, thrombocytopenia emerged as one of the major discrepant adverse effects that preferentially impacted the ≥70 cohort. However, a recent study of HR+, HER2− metastatic breast cancer patients receiving palbociclib-based therapy reported a similar prevalence of adverse effects to treatment and minimal differences in prevalence of specific adverse effects such as neutropenia, diarrhea, and fatigue between patients <70 and patients ≥70 over the course of the first six months of treatment [48]. Another study assessed the difference in prevalence of specific adverse effects associated with palbociclib, ribociclib, and abemaciclib and found neutropenia to be more prevalent in patients receiving palbociclib-based therapy and diarrhea to be more prevalent in patients receiving abemaciclib-based therapy [49]. Thus, nuances in the duration of treatment and the specific CDK4/6 inhibitor used may be responsible for discrepant findings regarding adverse effects and must be considered when assessing treatment-related toxicities.

One nuance worth noting in the dataset is that patients that received CDK4/6 inhibitor-based therapy combined with FUL almost invariably had received prior treatment with AI, whereas patients who received CDK4/6 inhibitor-based therapy with AI had often not received endocrine therapy for a year or greater prior to receiving AI. This key difference accounts, in part, for the differences in the duration of PFS and OS observed between the AI and FUL-treated groups throughout the study. Finally, since CDK4/6 inhibitor-based therapy is not curative, future work would be well-suited to build on recent studies of novel combinatorial strategies involving the inclusion of PI3K inhibitors and other novel agents with existing CDK4/6 inhibitor and endocrine therapy-based approaches [50,51,52,53]. The findings of this study can contribute to that end. 

A limitation of many previous studies has been the underrepresentation of older populations despite these individuals representing a large percentage of newly diagnosed breast cancer cases [19,54]. In this study, 34.7% of study participants were ≥70 years of age, which is far more representative of the actual percentage of individuals with metastatic breast cancer from that age group (~30%) [20,55]. Findings from this study can assist older populations weigh treatment options in terms of the tradeoffs between efficacy and tolerability of adverse effects that can further diminish quality of life, which, particularly in older patient populations, can be just as important as simply prolonging survival [56].

## 5. Conclusions

Here, we report that older populations, particularly individuals 70 years of age and older, display prolonged duration of PFS but not OS, when treated with CDK4/6 inhibitor-based therapy. However, the aged cohort presents more often with adverse effects to treatment, namely fatigue, diarrhea, and thrombocytopenia. Finally, findings from age group analysis identified the SBR score as a potential driver of PFS duration. Consistencies and discrepancies between the findings of this study and previous work highlight the nuances of specific CDK4/6 inhibitor responses, the impact of treatment duration, and the need for further investigation to help inform treatment decisions in older populations with metastatic breast cancer.

## Figures and Tables

**Figure 1 cancers-16-01749-f001:**
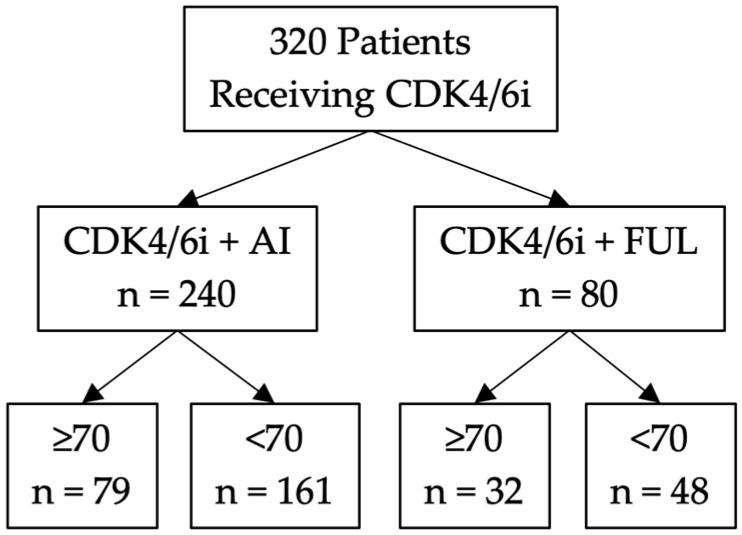
Elderly study patient population stratified by age. Consort diagram for the study of breast cancer patients binned by age (less than or at least 70 years of age) at Roswell Park Comprehensive Cancer Center receiving CDK4/6 inhibitor in combination with either an aromatase inhibitor or fulvestrant. CDK4/6i: cyclin-dependent kinase 4/6 inhibitor; AI: aromatase inhibitor; FUL: fulvestrant.

**Figure 2 cancers-16-01749-f002:**
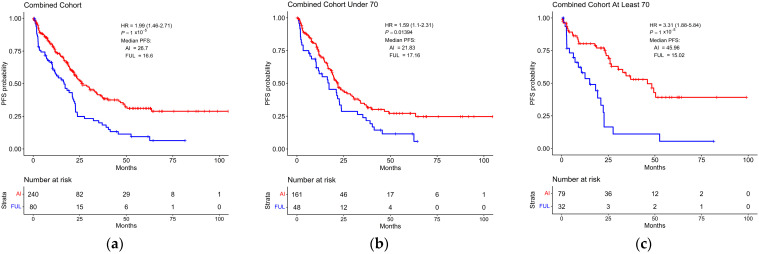
Differences in progression-free survival of patients taking either aromatase inhibitor or fulvestrant in combination with CDK4/6 inhibitor. (**a**) Kaplan–Meier plot displaying the duration of progression-free survival (PFS) for all patients receiving CDK4/6 inhibitor in combination with either aromatase inhibitor (AI) or fulvestrant (FUL). (**b**) Kaplan–Meier plot displaying the duration of PFS for patients under 70 years of age receiving CDK4/6 inhibitor in combination with either AI or FUL. (**c**) Kaplan–Meier plot displaying the duration of PFS for patients at least 70 years of age receiving CDK4/6 inhibitor in combination with either AI or FUL. HR: hazard ratio.

**Figure 3 cancers-16-01749-f003:**
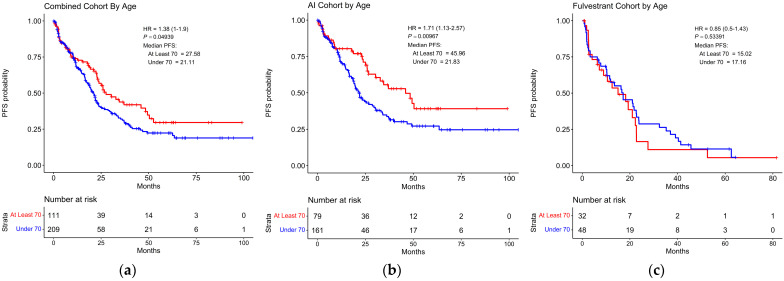
Differences in progression-free survival of patients under 70 years of age and at least 70 years of age taking either aromatase inhibitor or fulvestrant in combination with CDK4/6 inhibitor. (**a**) Kaplan–Meier plot displaying the duration of progression-free survival (PFS) for all patients 70 years of age and older or under 70 years of age receiving CDK4/6 inhibitor-based therapy. (**b**) Kaplan–Meier plot displaying the duration of PFS for patients 70 years of age and older or under 70 years of age receiving CDK4/6 inhibitor therapy in combination with an aromatase inhibitor. (**c**) Kaplan–Meier plot displaying the duration of PFS for patients 70 years of age and older or under 70 years of age receiving CDK4/6 inhibitor therapy in combination with fulvestrant. HR: hazard ratio.

**Table 1 cancers-16-01749-t001:** Prevalence of breast cancer subtype in each cohort.

	CDK4/6i + AI		CDK4/6i + FUL	
	<70 (n = 70)	≥70 (n = 45)	<70 (n = 24)	≥70 (n = 12)
Basal	1 (1.4)	1 (2.2)	0 (0.0)	0 (0.0)
HER2	1 (1.4)	2 (4.4)	4 (16.7)	2 (16.7)
LumA	32 (45.7)	29 (64.4)	13 (54.2)	3 (25.0)
LumB	34 (48.6)	13 (28.9)	6 (25.0)	6 (50.0)
Normal	2 (2.9)	0 (0.0)	1 (4.2)	1 (8.3)

Parentheses denote percent. CDK4/6i, CDK4/6 inhibitor; AI, aromatase inhibitor; FUL, fulvestrant; HER2, human epidermal growth factor receptor 2; LumA, luminal A; LumB, luminal B.

**Table 2 cancers-16-01749-t002:** Clinicopathological variates and their contribution to PFS as stratified by age.

	All Patients (n = 320) (<70 = 209; ≥70 = 111)				CDK4/6i + AI Patients (n = 240) (<70 = 161; ≥70 = 79)				CDK4/6i + FUL Patients (n = 80) (<70 = 48; ≥70 = 32)			
≥70 vs. <70	<70	≥70	HR	*p* val.	<70	≥70	HR	*p* val.	<70	≥70	HR	*p* val.
HER2 0+	37 (17.7)	15 (13.5)	1.34	0.6	33 (20.5)	12 (15.2)	1.17	0.8	4 (8.3)	3 (9.4)	2.40	0.4
HER2 1+	73 (34.9)	40 (36.0)	1.55	0.1	54 (33.5)	30 (38.0)	2.20	0.04	19 (39.6)	10 (31.3)	0.63	0.3
HER2 2+	74 (35.4)	41 (36.9)	1.36	0.2	56 (34.8)	32 (40.5)	1.57	0.1	18 (37.5)	9 (28.1)	0.70	0.4
PR−	74 (35.4)	42 (37.8)	1.36	0.2	50 (31.1)	28 (35.4)	1.64	0.1	24 (50)	14 (43.8)	0.92	0.8
PR+	115 (55.0)	56 (50.5)	1.50	0.1	98 (60.9)	48 (60.8)	1.78	0.05	17 (35.4)	8 (25.0)	0.69	0.4
SBR1	20 (9.6)	15 (13.5)	0.97	1	12 (7.5)	11 (13.9)	0.56	0.5	8 (16.7)	4 (12.5)	2.13	0.3
SBR2	112 (53.6)	66 (59.5)	1.29	0.2	85 (52.8)	45 (57.0)	1.77	0.04	27 (56.3)	21 (65.6)	0.73	0.4
SBR3	59 (28.2)	18 (16.2)	1.14	0.7	48 (29.8)	13 (16.5)	1.26	0.6	11 (22.9)	5 (15.6)	0.68	0.7
Non-Visceral	119 (56.9)	49 (44.1)	1.52	0.1	97 (60.2)	38 (48.1)	1.80	0.06	22 (45.8)	11 (34.4)	0.94	0.9
Visceral	90 (43.1)	62 (55.9)	1.38	0.1	64 (39.8)	41 (51.9)	1.74	0.05	26 (54.2)	21 (65.6)	0.83	0.6
Distant Mets	195 (93.3)	110 (99.1)	1.45	0.02	152 (94.4)	78 (98.7)	1.79	0.005	43 (89.6)	32 (100.0)	0.93	0.8
Local Mets	14 (6.7)	1 (0.9)	-	-	9 (5.6)	1 (1.3)	-	-	5 (10.4)	0 (0.0)	-	-
De novo Mets	74 (35.4)	29 (26.1)	1.31	0.41	66 (41.0)	27 (34.2)	1.34	0.43	8 (16.7)	2 (6.3)	0.39	0.31
Recurrent Mets	135 (64.6)	82 (73.9)	1.45	0.05	95 (59.0)	52 (65.8)	2.00	0.007	40 (83.3)	30 (93.8)	0.87	0.63
Prior ET	133 (63.6)	76 (68.5)	1.20	0.3	89 (55.3)	45 (57.0)	1.50	0.1	44 (91.7)	31 (96.9)	0.96	0.9
No Prior ET	76 (36.4)	35 (31.5)	1.86	0.06	72 (44.7)	34 (43.0)	1.95	0.05	4 (8.3)	1 (3.1)	-	-

Parentheses denote percent. CDK4/6i, CDK4/6 inhibitor; AI: aromatase inhibitor; FUL: fulvestrant; HR: hazard ratio; *p* val: *p* value; HER2: human epidermal growth factor receptor 2; PR: progesterone receptor; SBR: Scarff–Bloom–Richardson; Mets: metastases; ET: endocrine therapy.

**Table 3 cancers-16-01749-t003:** Adverse effects were reported by each cohort during treatment.

	CDK4/6i + AI		CDK4/6i + FUL	
	<70 (n = 161)	≥70 (n = 79)	<70 (n = 48)	≥70 (n = 32)
AE leading to dose reduction and/or hold	79 (49.1)	48 (60.8)	23 (47.9)	18 (56.3)
AE leading to discontinuation	9 (5.6)	6 (7.6)	0 (0.0)	5 (15.6)
Fatigue	66 (41.0)	43 (54.4)	16 (33.3)	17 (53.1)
Diarrhea	28 (17.4)	20 (25.3)	9 (18.8)	8 (25.0)
Leukopenia/Neutropenia	84 (52.2)	48 (60.8)	25 (52.1)	15 (46.9)
Thrombocytopenia	16 (9.9)	17 (21.5)	2 (4.2)	7 (21.9)
Anemia	38 (23.6)	28 (35.4)	8 (16.7)	9 (18.8)
Abnormal ALT/AST	11 (6.8)	3 (3.8)	1 (2.1)	1 (3.1)
Nausea/Vomiting	53 (32.9)	19 (24.1)	19 (39.6)	9 (18.8)

Parentheses denote percent. CDK4/6i, CDK4/6 inhibitor; AI: aromatase inhibitor; FUL: fulvestrant; AE: adverse event; ALT: alanine transaminase; AST: aspartate aminotransferase.

## Data Availability

All data reported in this study are available from the corresponding author upon reasonable request.

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
