# Peer review of "Real-World Experience among Elderly Metastatic Breast Cancer Patients Treated with CDK4/6 Inhibitor-Based Therapy"

_cancers, 2024, doi:10.3390/cancers16091749_

Round 1

Reviewer 1 Report

Comments and Suggestions for Authors

The manuscript "Patients ≥ 70 years of age with metastatic breast cancer display 2 prolonged progression-free survival when treated with CDK4/6 3 inhibitor-based therapy" describes results of a clinical trial that assessed the impact of CDK4/6i + fulvestrant/aromatase inhibitor - combined therapy. The paper is well-written, and the results are potentially relevant to the field. 

Considering the quality of the manuscript, I believe that the paper can be accepted for publication in its current form.

Author Response

We appreciate Reviewer 1's entirely positive feedback regarding our manuscript.

Reviewer 2 Report

Comments and Suggestions for Authors

1. Upload better quality image for figure 1

2.. There is not enough data to conclude if older patients would benefit from CDK4/6 inhibitors. Especially since this is only in PFS and not OS. Therefore, this could be a spurious correlation more than any significant finding. 

Since all patients were on CDK4/6 inhibitors, it is inaccurate to conclude that older populations display improved efficacy (due to higher PFS compared to younger patients) "when treated with CDK4/6 inhibitors". The comparison is incomplete without older patient group without CDK4/6 inhibitor treatment. Please re-phrase this in conclusion as well as the title. The title suggests using CDK4/6i helps older patients. Make clear that this is only in comparison with younger patients. 

Author Response

The introduction must be improved.

We appreciate the constructive feedback from the reviewer that the introduction needed to be improved. Thus, we provided more background information to establish the rationale for the study, the study design, and the knowledge gaps of the field that we address in our study. In the process, we added a significant number of references to our revised introduction.

The results can be improved.

We appreciate the constructive feedback from the reviewer that the results could be improved. Thus, we separated the results into individual subsections for better readability. We also directly addressed the main clinicopathological variates that were found to be significantly different between the two age cohorts in their contribution to PFS and included two more reference studies that validated our findings that the older patient population unsurprisingly experiences a higher frequency of adverse effects leading to dose reductions, holds, or discontinuations when treated with CDK4/6 inhibitor-based therapy.

The conclusions must be improved.

We appreciate the constructive feedback from the reviewer. We have made significant additions to the discussion and conclusions sections in order to properly highlight the current knowledge gaps within the field, compare our findings to the findings of other groups, to note the limitations of our study, and to highlight future areas of inquiry. To this end, we have more than doubled the studies cited in our revised manuscript.

Upload a higher quality image for figure 1.

We appreciate the reviewer pointing out the issue with the display of Figure 1 and this issue has been corrected.

There is not enough data to conclude if older patients would benefit from CDK4/6 inhibitors. Especially since this is only in PFS and not OS. Therefore, this could be a spurious correlation more than any significant finding. Since all patients were on CDK4/6 inhibitors, it is inaccurate to conclude that older populations display improved efficacy (due to higher PFS compared to younger patients) "when treated with CDK4/6 inhibitors". The comparison is incomplete without older patient group without CDK4/6 inhibitor treatment. Please re-phrase this in conclusion as well as the title. The title suggests using CDK4/6i helps older patients. Make clear that this is only in comparison with younger patients. 

We agree that the statements regarding prolonged PFS observed in the older cohort receiving CDK4/6i plus AI as compared to the younger cohort receiving the same treatment need clarification of the comparisons being made. Thus, we have adjusted the title of the study from “Patients ≥ 70 years of age with metastatic breast cancer display prolonged progression-free survival when treated with CDK4/6 inhibitor-based therapy” to “Real-World Experience Among Elderly Metastatic Breast Cancer Patients Treated with CDK4/6 Inhibitor-Based Therapy.”

Reviewer 3 Report

Comments and Suggestions for Authors

The abstract is good but needs to be quantified. The Introduction section may include the gap analysis and limitations also. The statistical part is to be improved. The discussion section will be modified with respect to the overall research. The methodology is fine, however it requires improvement. The reference papers are to be enhanced. The conclusion needs to be modified. Comparison of previous and related works are to be analyzed in the clinical context.

Author Response

The abstract needs to be quantified.

We appreciate the reviewer pointing out this shortcoming of our initial submission and have included a quantified abstract for the resubmission.

The introduction may include gap analysis and limitations.

We appreciate the constructive feedback from the reviewer that the introduction needed to be improved. Thus, we provided more background to establish the rationale for the study, the study design, and the knowledge gaps of the field that we address in our study. In the process, we added a significant number of references to our revised introduction.

The methods and statistical analysis can be improved.

We appreciate the reviewer’s constructive feedback and have elaborated on the methodology significantly, including specific drugs used for treatment, the dates the study was conducted, identified that the study was conducted in accordance with the Declaration of Helsinki, included the specific statistical tests used and defined statistical significance cutoffs, rationale for study design using age 70 as the stratification point based on International Society for Geriatric Oncology recommendations, and the parameters used for custom R code data extraction.

The results can be improved.

We appreciate the constructive feedback from the reviewer that the results could be improved. Thus, we separated the results into individual subsections for better readability. We also directly addressed the main clinicopathological variates that were found to be significantly different between the two age cohorts in their contribution to PFS and included two more reference studies that validated our findings that the older patient population unsurprisingly experiences a higher frequency of adverse effects leading to dose reductions, holds, or discontinuations when treated with CDK4/6 inhibitor-based therapy.

The conclusions and discussion can be improved in regard to the overall research. Comparison of previous and related works are to be analyzed in the clinical context.

We appreciate the constructive feedback from the reviewer. We have made significant additions to the discussion and conclusions sections in order to properly highlight the current knowledge gaps within the field, compare our findings to the findings of other groups, to note the limitations of our study, and to highlight future areas of inquiry. To this end, we have more than doubled the studies cited in our revised manuscript.

The references can be improved.

We appreciate the reviewer bringing our attention to the lack of references in our initial submission. Our revised manuscript has increased the number of references from 20 to 56 pertinent studies that add better perspective to our work and the knowledge gaps within the field that our study addresses.

Round 2

Reviewer 2 Report

Comments and Suggestions for Authors

Thank you for the edits. Manuscript looks ready.